# Sensitive Coatings Based on Molecular-Imprinted Polymers for Triazine Pesticides’ Detection [note 1]

**DOI:** 10.3390/s24185934

**Published:** 2024-09-13

**Authors:** Usman Latif, Sadaf Yaqub, Franz L. Dickert

**Affiliations:** 1Department of Analytical Chemistry, University of Vienna, Waehringer Str. 38, A-1090 Vienna, Austria; 2Interdisciplinary Research Centre in Biomedical Materials, COMSATS University Islamabad-Lahore Campus, Lahore 54600, Pakistan

**Keywords:** pesticide, pesticide detection, molecular imprinting, MIP, quartz crystal microbalance, sensor, acid-to-cross-linker ratio, supramolecular chemistry

## Abstract

Triazine pesticide (atrazine and its derivatives) detection sensors have been developed to thoroughly check for the presence of these chemicals and ultimately prevent their exposure to humans. Sensitive coatings were designed by utilizing molecular imprinting technology, which aims to create artificial receptors for the detection of chlorotriazine pesticides with gravimetric transducers. Initially, imprinted polymers were developed, using acrylate and methacrylate monomers containing hydrophilic and hydrophobic side chains, specifically for atrazine, which shares a basic heterocyclic triazine structure with its structural analogs. By adjusting the ratio of the acid to the cross-linker and introducing acrylate ester as a copolymer, optimal non-covalent interactions were achieved with the hydrophobic core of triazine molecules and their amino groups. A maximum sensor response of 546 Hz (frequency shift/layer height equal to 87.36) was observed for a sensitive coating composed of 46% methacrylic acid and 54% ethylene glycol dimethacrylate, with a demonstrated layer height of 250 nm (6.25 kHz). The molecularly imprinted copolymer demonstrated fully reversible sensor responses, not only for atrazine but also for its metabolites, like des-ethyl atrazine, and structural analogs, such as propazine and terbuthylazine. The efficiency of modified molecularly imprinted polymers for targeted analytes was tested by combining them with a universally applicable quartz crystal microbalance transducer. The stable selectivity pattern of the developed sensor provides an excellent basis for a pattern recognition procedure.

## 1. Introduction

The US Environmental Protection Agency (EPA) defines a pesticide as any substance or combination of substances used for pest control, plant regulation, defoliation, or nitrogen stabilization [1]. Therefore, pesticides are utilized to safeguard crops from insects, weeds, fungi, and various pests, leading to enhanced crop yields. However, their misuse can lead to residue issues in crops, soil, and water, thus posing significant health concerns [2]. Certain pesticides infiltrate soils and underlying groundwater through direct application and later on enter the food chain [3]. Contaminant levels in food samples are rapidly increasing due to improper agricultural practices. Excessive use of pesticides is a major cause of contamination in agricultural crops [4]. Additionally, these pesticides are also used to manage pests and weeds in human habitats, animal farming, and during crop storage after harvesting [5]. Even at low doses, these pesticides exhibit severe toxic effects on epithelial, neural, immune, hepatic, and reproductive systems, and they are also known to possess carcinogenic, mutagenic, and teratogenic properties once absorbed into the body [6]. Notably, lipophilic pesticides can pass through breast milk to infants [7]. Exposure to pesticides and herbicides can lead to various health complications in humans, such as cancer [8], neurotoxicity [9], Alzheimer’s [10], Parkinson’s [11], infertility [12], leukemia [13], diabetes [14], and asthma [15]. Thorough checks for the presence of these toxic chemicals in food items should be conducted before consumption to prevent their exposure to humans.

The advancement of highly sensitive, selective, and fast sensors enables the on-site detection of pesticides in comparison to sophisticated and costly instruments, such as high-performance liquid chromatography (HPLC) and gas chromatography–mass spectrometry (GC-MS), which demand skilled operators and involve time-consuming sample preparation [16,17]. The adoption of bio- or chemo-sensor technologies capable of swiftly detecting pesticide residues is crucial and could serve as a cost-effective solution for monitoring pesticides in the environment [18]. Various biosensors have been developed to detect pesticides; however, their recognition capability is compromised in varying pH, temperature, and organic solvent conditions [19]. Molecularly imprinted polymers (MIPs) emerge as a promising alternative due to their stability, consistent performance in extreme conditions, and ease of production [20,21]. MIPs hold significant promise for selective recognition across various applications. However, their practical use is hindered by several limitations, including incomplete template removal, limited availability of templates, heterogeneity of binding sites, and issues with mechanical and chemical stability under harsh conditions. Additionally, slow kinetics and limited reusability can negatively impact the reproducibility and selectivity of MIPs. Porous polymers and suitable monomers can partially be a solution for these impediments.

The strong yet reversible interaction between basic chlorotriazines and the acidic nature of the polymeric chain is a critical factor in designing a sensitive coating for the detection of a target analyte (pesticide). In designing an MIP, using methacrylic acid (MAA) as the functional monomer and ethylene glycol dimethacrylate (EGDMA) as the cross-linker, the ratio of acid to cross-linker is a crucial factor that influences the polymer’s binding efficiency, selectivity, and mechanical properties. The optimal ratio depends on the nature of the template molecule and the strength of the interaction between MAA and the template. A higher amount of methacrylic acid can increase the number of available binding sites, whereas a higher proportion of EGDMA increases the cross-linking density, resulting in a more rigid and stable polymer matrix. This rigidity is beneficial for maintaining the shape, size, and integrity of the binding sites in the cavities formed. The choice of solvent and polymerization conditions also influences the optimal monomer-to-cross-linker ratio, as these factors affect the polymer’s porosity, surface area, and overall performance. Numerous studies in the literature have designed molecularly imprinted polymers (MIPs) by optimizing the MAA-to-EGDMA ratio [22,23,24,25]. Similarly, in this study, the acid-to-cross-linker ratio was optimized to develop a stable polymeric system for effective recognition as well as re-inclusion of the templated analyte. The combination of molecular imprinting polymers with quartz crystal microbalance (QCM) transducers enabled us to develop sensitive and selective sensors for detecting pesticides such as atrazine (ATR), as well as its metabolite des-ethyl atrazine (DEA) and structural analogs, e.g., propazine (PRO) and terbuthylazine (TBA). Thus, variations in the ratio of the acid (methacrylic acid) to the cross-linker (ethylene glycol dimethacrylate) result in an effective coating for QCM sensors. The chemical structures of atrazine and its metabolite and structural analogs are given in Figure 1.

## 2. Materials and Methods

### 2.1. Reagents

The monomers 1-vinyl-2-pyrrolidone (VP) and methacrylic acid (MAA); cross-linkers N,N-methylenebisacrylamide (MBA) and ethylene glycol dimethacrylate (EGDMA); triazine pesticides, such as ATR, DEA, PRO, and TBA; and all other reagents, such as THF, were purchased from Merck (Darmstadt, Germany).

### 2.2. Instrumentation

Quartz crystals (with a fundamental resonance frequency of 10 MHz) were purchased from Zhejiang Quartz Crystal Electronic Company, Shanghai, China. Dual-electrode geometries were screen-printed onto quartz discs with gold paste (brilliant gold paste from Heraeus) using a silk sieve (mesh size: 41 μm). This arrangement allowed us to measure MIP and non-imprinted polymer (NIP) responses separately [26]. At the same time, both frequency and temperature fluctuations were compensated for by adopting differential measurements. Mass-sensitive experiments were conducted by affixing the QCM sensor to a flow cell. The flow cell can accommodate a 150 μL volume of the test sample. We connected custom-built oscillator circuits to the dual-electrode QCM of the flow cell. The QCM electrodes were part of a feedback loop of these oscillators as frequency-determining elements. Any mass load on the oscillator leads to a frequency response. The flow cell was connected to a frequency counter (HP 53131A) to transfer data via a computer using a GPIB cable. The mass-sensitive sensors were exposed to sample solutions through a peristaltic pump operating at a flow rate of 1.5 mL/min. Surface profiling of sensitive coatings was conducted by utilizing an Atomic Force Microscope (Veeco Nanoscope IVa) in contact mode.

### 2.3. Designing Sensitive Coatings for Detecting Pesticides

At first, the sensitive coating for pesticide detection was designed by utilizing vinyl-pyrrolidone as the monomer, cross-linked with bisacrylamide. The existence of a strong interaction between the halogen atoms of the triazines and the highly polarizable aromatic ring of the polymer also renders it suitable for imprinting. A sensitive coating based on pyrrolidone was synthesized using 8.6 mg/L of pesticide as a template. This coating was then exposed to propazine concentrations of 2.1, 4.3, and 8.6 mg/L, with the highest response of 350 Hz observed at 8.6 mg/L of propazine. The sensor response was also very sluggish; that is why another sensitive coating was designed to enhance the sensitivity and responsiveness.

#### 2.3.1. Effect of Acid Content on Sensitivity of Sensor Coatings

Methacrylic acid was selected to design sensitive coatings for chlorotriazine after examining the sensor responses in initial experiments with diverse polymers. All polymer blends were synthesized with 2.5 mg of AIBN as the initiator in the THF and heated at 60 °C until the gel point was reached. A 10% pesticide by weight was utilized as a template for varying acid and cross-linker ratios. The pre-polymerized solutions (both imprinted and non-imprinted) were then applied onto separate electrodes of the dual-electrode QCM and exposed to UV light overnight for complete polymerization. The strong but reversible interaction between basic chlorotriazines and the acidic nature of a polymeric chain is a crucial determinant in selecting the suitable monomer. Moreover, the optimization of the acid (monomer)-to-cross-linker ratio of the polymeric chain is key to developing the most efficient recognition sites for the pesticide of interest. To achieve this, polymer mixtures were formulated by mixing MAA and EGDMA in different ratios. The frequency shifts and layer heights of these sensitive coatings were recorded, and they are presented in Table 1. All measurements were conducted at 28 °C due to lower analyte solubility at colder temperatures. Otherwise, minor sensor responses would have been obtained.

A sensitive coating containing 28% of MAA and 72% of EGDMA and with a layer height of 320 nm (1 kHz change in frequency is equal to 40 nm) furnishes a sensor response of 200 Hz (frequency shift/layer height is equal to 200/8 = 25). The sensitivity of the sensor is enhanced as the MAA ratio is increased, resulting in a stronger interaction between the acidic polymer matrix and the basic template molecules. A maximum sensor response of 546 Hz (frequency shift/layer height is equal to 87.36) was observed for a sensitive coating with 46% of MAA and 54% of EGDMA and a layer height of 250 nm (6.25 kHz). The ratio of acid to cross-linker was crucial for both the stability of the polymer matrix and the effective re-inclusion of the template molecules. Increasing the acid content beyond 46% led to a reduction in sensor response, attributed to inadequate cross-linking within the imprinted polymer matrix. Therefore, an optimal acid amount of around 46% was identified for maximizing sensitivity to a pesticide in designing sensor coatings.

All sensor responses in our experiment are attributed to bulk phenomena. This conclusion is supported by test measurements where the layer height was increased without modifying other parameters. To further validate our findings, we performed comparative measurements using uncoated gold electrodes. Additionally, we employed a thiol compound (octadecanethiol) to create a dense monolayer on the gold electrode surface. This particular thiol was selected due to its molecular weight being comparable to (even higher than) that of atrazine.

In our tests, the thiol produced a sensor response of 140 Hz, which was significantly lower than the 546 Hz response generated by atrazine. This difference highlights the higher sensitivity of the sensor to atrazine, confirming the bulk effect.

The observed maximum in the MAA/EGDMA ratio is driven by two opposing mechanisms. On the one hand, an increase in MAA enhances the sensor’s interaction with atrazine, extending the binding affinity. On the other hand, a reduction in EGDMA results in fewer cavities within the polymer matrix, impacting its structural characteristics and vice versa. Therefore, it can be inferred that a host–guest interaction mechanism plays a pivotal role in the sensor response observed in this system.

#### 2.3.2. Template Removal

Templated molecules that are loosely bound to imprinted polymers can easily be removed by washing or heating. However, in the case of strong interactions, the removal of templated moieties from imprinted polymers becomes exceedingly difficult. The removal of atrazine and related structural analogs from imprinted polymers by washing with water is difficult because of the strong interaction of the polymeric structure with triazine moieties. The efficiency of different solvents (water, methanol, acetic acid, or an acetic acid and methanol mixture) was assessed for removing triazine moieties from polymeric structures. A good template removal solvent should possess the following properties: (1) It should completely remove the templated molecules. (2) It should not deform or disturb the structural features of the recognitions sites or the polymer. (3) It should not leach out of the polymer. In our case, IR spectroscopy was used to analyze the removal of the templated molecules from the polymeric matrix. The sensitivity of the sensor remains the same even after multiple washings and reuses. Additionally, the sensor frequency returns to its initial value after each washing. A general observation is that the non-imprinted layers are more robust compared to the imprinted layers and show a relatively less reduction in mass, which can be observed from the frequency shifts after each experiment (washing).

The efficiency of methanol as a template removal solvent was also investigated based on the time required for the removal of the template from different polymer coatings. The amount of template removal time varies with the change in the nature or composition of the polymer layer. Methanol was selected as the best template removal solvent because it shows higher template removal behavior, less mass-loss, and sustainable sensor response over multiple washings/uses. A network analyzer was used to record the mass-loss after each washing. Improved sensitivity was observed when using methanol as the template removal, with a greater number of recognition sites available for the re-inclusion of the templated molecules in acrylate sensor coatings.

#### 2.3.3. IR Analysis for Template Removal

Attenuated Total Reflectance Infrared (ATR-IR) spectroscopy is a valuable technique used to analyze the removal of template molecules from molecularly imprinted polymers (MIPs). This analysis helps confirm the successful extraction of the template from the polymer matrix, ensuring the formation of specific recognition sites for the target analyte. Infrared studies were utilized to confirm atrazine removal from imprinted polymer film. All spectra were obtained using the FT-IR spectrometer (Perkin-Elmer, Inc., Shelton, CT, USA) equipped with ZnSe crystal. Imprinted and non-imprinted polymer films were coated on quartz glass for the FTIR studies (background spectra was collected using quartz glass).

Figure 2 displays the ATR-IR spectra of both the imprinted and reference polymers before and after washing with methanol. The IR spectra of the imprinted polymer revealed the absence of the amide band at 3270 cm^−1^ (after washing with methanol). Notably, the absorptions for the symmetric stretching of –CH_3_ and –CH_2_, as well as the deformations of the alkyl group, remained consistent across all cases.

The surface of the imprinted layer coated on the QCM electrode was examined for surface roughness using an Atomic Force Microscope (Veeco Instruments Inc., Fullerton, CA, USA). Figure 3 displays the image captured via AFM. The AFM image of the molecularly imprinted polymer coating was taken in contact mode with a silicon nitride tip, which revealed the surface morphology of the pesticide recognition layer. TEM is the most powerful tool among microscopic techniques [27]; however, in this study, AFM was preferred over SEM or TEM for surface profiling because surface characteristics can be analyzed at an ambient temperature [20]. Additionally, the thickness of the sensor layer was assessed using a network analyzer, and no damage to the thin film was detected.

## 3. Results and Discussion

An acrylate coating for detecting atrazine was synthesized by using methacrylic acid as the monomer, ethylene glycol dimethacrylate as the cross-linker, and AIBN as the initiator. A pre-polymer was prepared by adding 46% methacrylic acid, 54% ethylene glycol dimethacrylate by weight, 10% atrazine as the template (in 1 mL of THF), and a drop of AIBN as the initiator. The pre-polymerized solution was spin-coated on the dual electrodes of the QCM sensor, followed by overnight polymerization under a UV lamp in a nitrogen atmosphere. The template molecules were washed out by keeping the QCM sensor in methanol for 30 min. A corresponding reference coating was also prepared using a similar procedure but without pesticide. The responses of the reference (375 nm thickness) and imprinted polymer (480 nm thickness) layers on electrodes were monitored by gravimetric analysis.

The efficacy of the sensitive coating was analyzed by exposing it to a pesticide solution, and a concentration-dependent response was observed, as shown in Figure 4. For this purpose, a concentrated solution (7 mg/L) of pesticide was prepared and diluted further to analyze the sensor response to other, lower concentrations. The flow cell (containing the QCM sensor) was flushed with distilled water at a rate of 1.5 mL/min by using a peristaltic pump to establish equilibrium before starting the gravimetric measurements. To maintain thermal stability, both the flowing water and sample solutions were kept at 28 °C during the experiment. A sensor response was recorded for the 7 mg/L atrazine solution using the imprinted polymethacrylic acid layer. The reference electrode also exhibited a frequency shift of around 100 Hz, potentially due to the surface absorption of analyte molecules by the non-imprinted polymer. Further investigations into the sensitivity of the imprinted film were conducted using other dilutions of atrazine, extending down to 0.35 mg/L.

Figure 5 depicts the sensitivity of a PRO-imprinted sensitive coating composed of methacrylic acid and ethylene glycol dimethacrylate. A concentration-dependent response was observed while exposing the PRO-imprinted sensitive layer to different concentrations of PRO. The sensor response was lower for PRO in comparison to the response of the atrazine-imprinted sensitive coating and other structural analogs to their templated analytes. The presence of bulky groups on both ends of the triazine ring diminishes the basic nature of PRO. As a result, the weak interactions between the less basic triazine (PRO) and the acid moieties of the polymer chain led to minor mass changes and decreased frequency shifts.

Afterward, this polymer was imprinted with TBA, and the sensitivity of TBA-imprinted sensor coatings was analyzed by exposing them to different concentrations (0.35 mg/L to 7 mg/L) of a templated analyte. Concentration-dependent frequency shifts were observed while exposing the TBA-imprinted sensitive coatings to different concentrations of TBA, as shown in Figure 6.

The graph clearly shows a concentration-dependent sensor response; pronounced frequency shifts were observed with an increase in analyte concentrations.

In the next experiment, the polymer was imprinted with DEA and exposed to different concentrations (0.35 mg/L to 7 mg/L) of the templated analyte. The concentration-dependent frequency shifts of the sensor are shown in Figure 7. The most significant resonance frequency shift of the patterned sensitive coating was observed for DEA, with a frequency shift of 600 Hz at 7 ppm (Figure 7), in comparison to atrazine, terbuthylazine, and propazine. The higher-frequency shift was observed due to the pronounced basic nature of the pesticide and the strength of its interaction with the imprinted layer of polymethacrylic acid. The higher sensitivity of polymethacrylic acid to DEA (at 7 ppm) was also attributed to the relatively easy access of the analytes to the recognition sites because of less steric hindrance. Removing the ethyl group from atrazine to form des-ethyl atrazine, increases its basic nature, resulting in a higher sensor response because of the strong interaction (hydrogen bonding) of DEA with MIPs.

It can be seen in the figure that the limiting concentration is less than 0.1 mg/L (signal/noise ratio ≥3). The base unit of triazine herbicides (ATR, PRO, TBA, DEA, and their chlorinated metabolites) is the same, and they have very minute differences in their structure based on the groups attached to the base unit. Designing a sensitive coating that can selectively bind the targeted pesticide is definitely a difficult task. However, this was achieved by applying principles of supramolecular chemistry, especially through molecular imprinting. This imprinting process creates patterned polymers designed for the re-inclusion of pesticides. It is evident that molecular imprinting for propazine will create selective interaction sites for selectively binding the propazine molecules, but one would expect that it will also interact with other triazine metabolites.

To analyze the cross selectivity, the four sensitive coatings were prepared by imprinting PRO, TBA, ATR, and DEA. All these sensors were exposed to all four templated analytes, and the responses of these sensitive coatings are shown in Figure 8.

These sensor responses were normalized before comparison. Each sensitive coating is selective to its templated analyte, except PRO, whereas in the case of DEA, the sensor sensitivity is higher to its templated analyte in comparison to the sensitivity of other coatings (PRO, TBA, ATR). Their heightened sensitivity is ascribed to their inherently pronounced basic nature. The metabolites of atrazine, i.e., des-isopropyl atrazine (DIA) and DEA, have a pronounced basic nature and, thus, additional sites available for hydrogen bonding with the acidic moieties of a polymer chain. These features enable DEA-imprinted coatings to show higher sensitivity in comparison to other coatings. On the other hand, the steric effect of bulky alkyl groups in propazine and terbuthylazine hinders the hydrogen bonding and therefore decreases their basic nature, which ultimately reduces their sensitivity even though the chemical nature of the polymer chain is same for each analyte. Obviously, hydrogen bonding between -COOH and aliphatic amines is the primary reason of interaction. This interaction is reduced by steric substituents; however, the derivatives DEA and ATR show pronounced responses. The consistent selectivity pattern over time with the same sensor is an excellent platform for a pattern recognition procedure [28,29].

There have been several reports in the literature regarding atrazine detection using various recognition entities and transducers. However, we have focused on studies that relied on imprinting technology to detect atrazine. The sensitivity of the developed detection system to atrazine, propazine, terbuthylazine, and des-ethyl atrazine is on par with other leading sensors, showcasing the effectiveness of imprinting technology, as shown in the following Table 2. The results presented are difficult to compare because most sensor responses are influenced by bulk effects, which depend on the layer height. An increased coating height enhances sensitivity but also causes a slower sensor response. Furthermore, nearly all strategies exhibit some temperature dependency, which can cause issues at very low concentrations because these fluctuations may limit the detection capability. 

Table 2 includes six recent MIP papers from the literature representing favorable transducer principles. It begins with the fluorescence of quantum dots, which generally exhibit high sensitivity [30]. Salahshoor et al. [31] discuss another optical strategy based on reflectometric interference spectroscopy. Agarwal et al. [32] achieved a highly sensitive optical sensor using the surface plasmon resonance (SPR) concept. Surface-enhanced Raman spectroscopy (SERS), which requires considerable investment in equipment but offers high sensitivity and structural data, is presented by Zhao et al. [33]. Potentiometric methods can also be applied to MIP coatings, as cited in the literature [34]. Cyclic voltammetry (CV) is frequently used for analyte detection by MIPs, with atrazine, a redox-active compound, analyzed by Lattach et al. using this method [35].

Our results are listed as number 7. Sensitivity can be enhanced by utilizing nanoparticles due to their higher surface area, as demonstrated in our earlier paper [20]. Sensor coating patterning, previously carried out by Park [36], yielded results surpassing those of the earlier study [37]. Additionally, the literature [21,38,39,40,41] provides further information related to the studies already listed in Table 2.

## 4. Conclusions

This study focuses on creating artificial receptors using a molecular imprinting technique to detect chlorotriazine pesticides through the self-organization of monomers around template species, resulting in highly selective molecular imprints at the nanometer and sub-nanometer scale. Imprinted polymers incorporating both hydrophilic and hydrophobic side chains from acrylate/methacrylate monomers were developed for atrazine. The structural analogs of atrazine share its fundamental heterocyclic triazine structure.

Fabricated sensor devices exhibit selective and reversible responses to their templated analytes when exposed to a concentrations ranging from 0.35 mg/L to 7 mg/L, with a limit of detection (LOD) varying between 0.1 to 0.2 mg/L. The optimized sensitive coating composition includes 46% methacrylic acid and 54% ethylene glycol dimethacrylate, with a layer height of 250 nm. The metabolites of atrazine, namely des-isopropyl atrazine (DIA) and DEA, have a pronounced basic nature, providing additional sites for hydrogen bonding with the acidic moieties of the polymer chain. These characteristics enable DEA-imprinted coatings to exhibit higher sensitivity compared to other coatings. Mass-sensitive detection, particularly in the context of sensors such as QCM or Surface Acoustic Wave (SAW) devices, offers the advantage that responses are directly related to frequency. Devices operating at higher frequencies, including the GHz range, could enhance the LOD in a significant way.

## Figures and Tables

**Figure 1 sensors-24-05934-f001:**
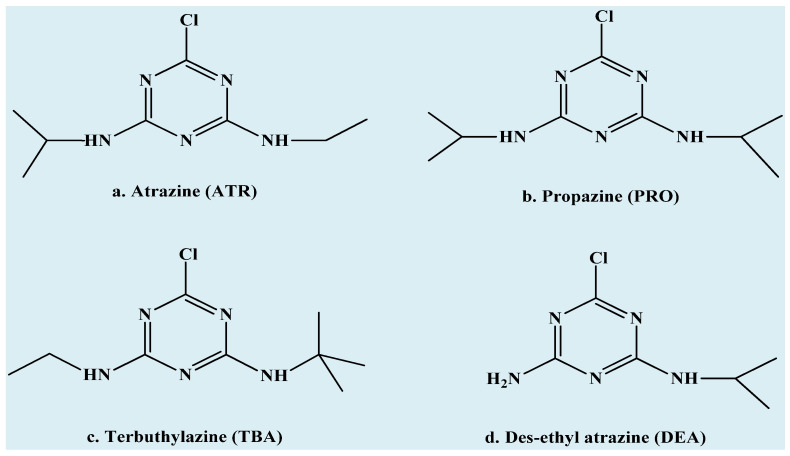
Chemical structures of triazine pesticides: (**a**) ATR, (**b**) PRO, (**c**) TBA, and (**d**) DEA.

**Figure 2 sensors-24-05934-f002:**
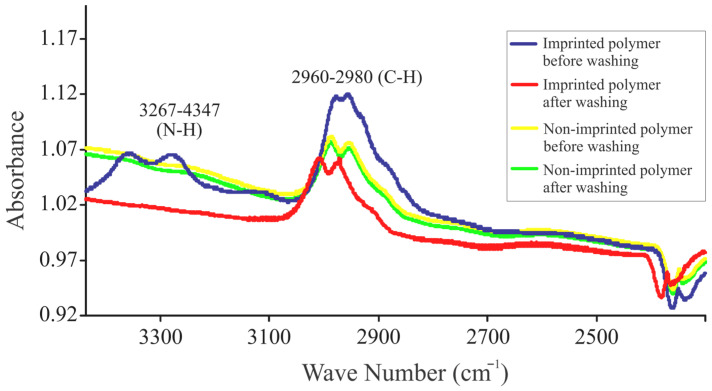
ATR-IR spectra of atrazine imprinted and reference polymers before and after washing with methanol.

**Figure 3 sensors-24-05934-f003:**
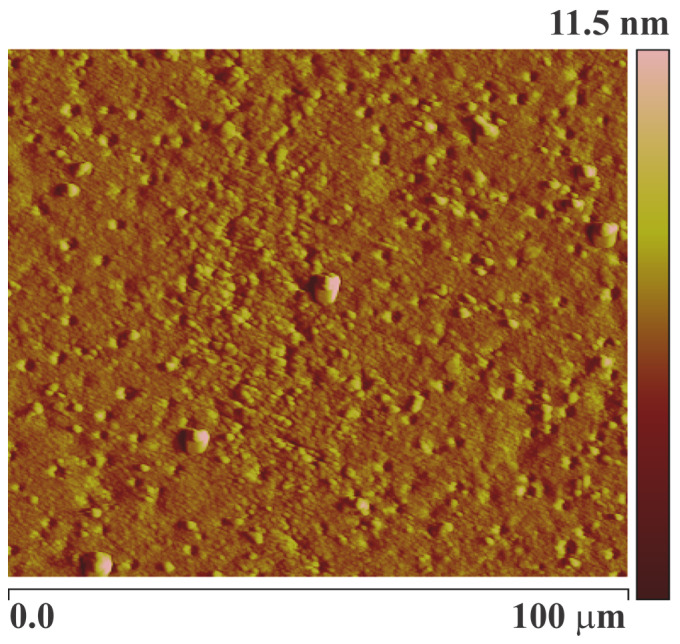
AFM of pesticide-imprinted acrylate coating. Methacrylic acid (MAA) as the monomer, ethylene glycol dimethacrylate (EGDMA) as the cross-linker, and atrazine as the template were used to prepare the atrazine-imprinted polymer.

**Figure 4 sensors-24-05934-f004:**
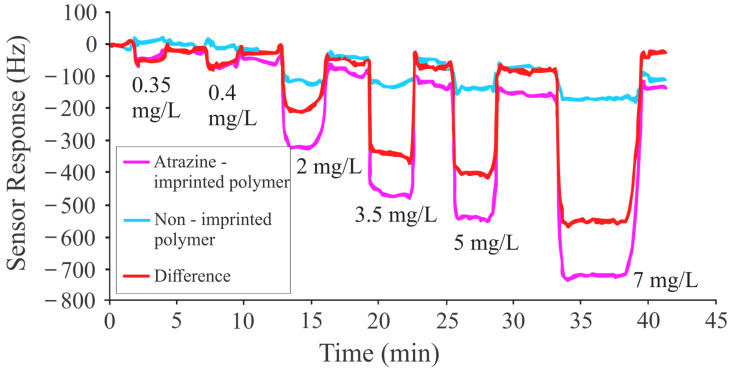
Sensor response of atrazine-imprinted polymer layer to different concentrations of the templated analyte.

**Figure 5 sensors-24-05934-f005:**
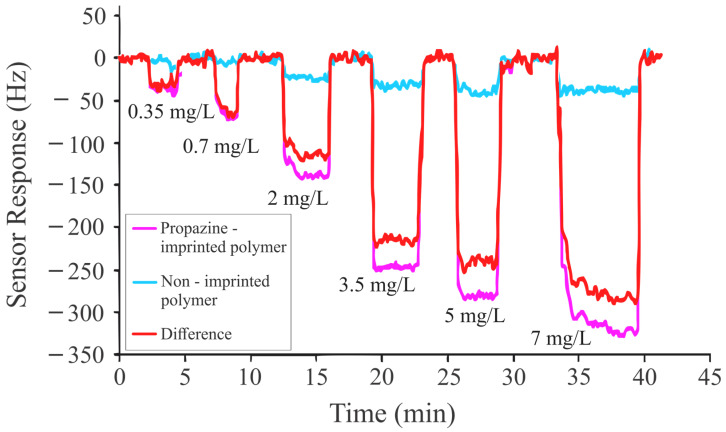
Sensor response of PRO-imprinted coating against different concentrations of propazine ranging from 0.35 mg/L to 7 mg/L.

**Figure 6 sensors-24-05934-f006:**
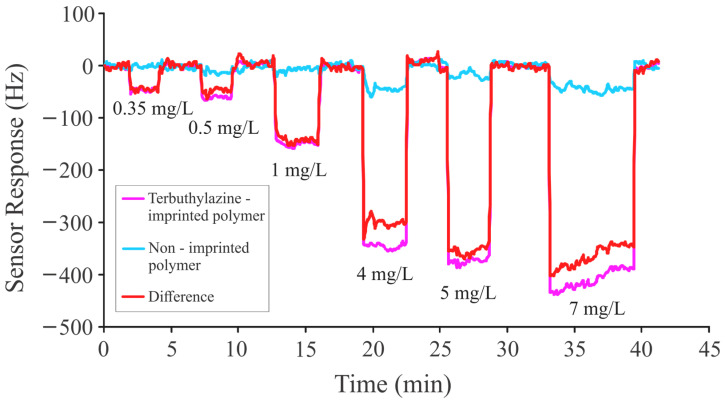
Sensor responses of terbuthylazine-imprinted coatings to different concentrations of TBA ranging from 0.35 mg/L to 7 mg/L.

**Figure 7 sensors-24-05934-f007:**
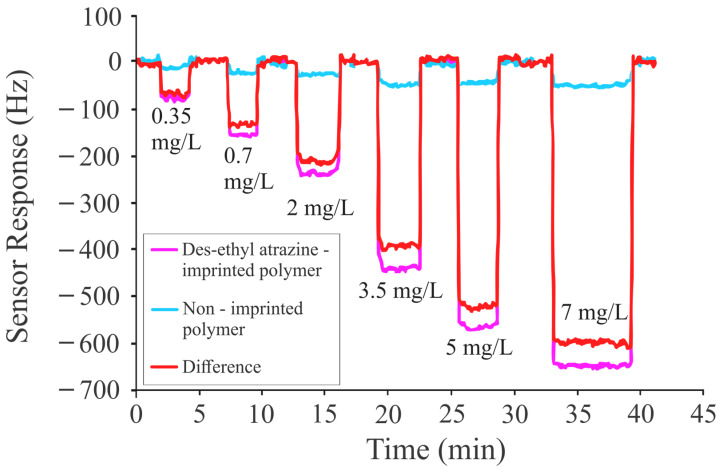
Sensor responses of DEA-imprinted coatings against different concentrations of DEA ranging from 0.35 mg/L to 7 mg/L.

**Figure 8 sensors-24-05934-f008:**
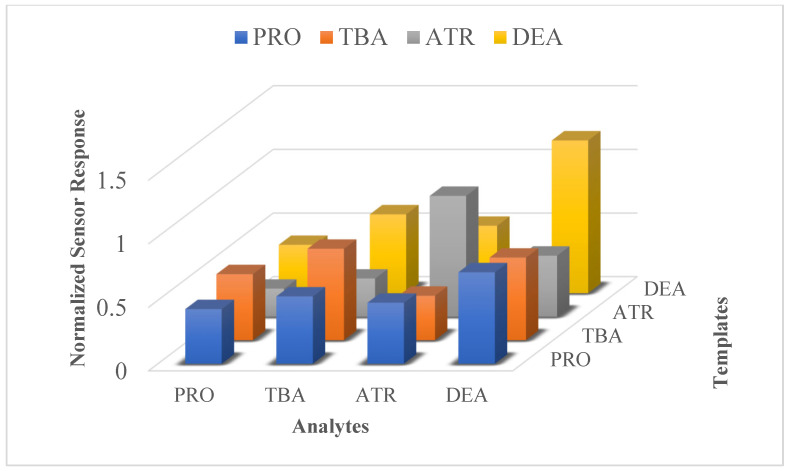
Cross-sensitivity responses of pesticide-imprinted coatings of PRO, TBA, ATR, and DEA against their templated analytes and interfering species. Frequency shifts were normalized for layer thickness, and frequency shifts of reference electrodes were subtracted in each case.

**Table 1 sensors-24-05934-t001:** The table includes the compositions of polymethacrylate used to optimize the monomer (methacrylic acid)-to-cross-linker (ethylene glycol dimethacrylate) ratio, along with their corresponding frequency shifts and layer heights to 7 mg/L atrazine in water.

MAA:EGDMA	Layer Height (kHz)	Net Frequency Shift (Hz)
28:72	8	200
30:70	10	354
33:67	6	198
35:65	5.2	314
40:60	5	408
46:54	6.25	546
50:50	4	180
60:40	4.2	84

**Table 2 sensors-24-05934-t002:** This table compares the results of those studies that rely on imprinting technology to detect atrazine.

Sr. No.	Recognition Element	Pesticide	Detection Method	Detection Range	LOD	Ref
1	CdSeTe/ZnS (QD)@MIP	Atrazine	Fluorescence	2–20 mol/L	0.8 × 10^−7^ mol/L	[30]
2	Acrylic acid, ethylene glycol dimethacrylate	ATR, DEA and de-isopropyl atrazine (DIA)	Photonic MIP	0.1 to 10 ppb	0.1, 0.2, and 0.3 ppb for ATR, DEA, and DIA, respectively	[31]
3	2-hydroxyethyl methacrylate, Ethylene glycol dimethacrylate	Atrazine	Surface plasmon resonance	10^−12^ to 10^−7^ M	1.92 × 10^−14^ M	[32]
4	Methacrylic acid, ethylene glycol dimethacrylate and AuNPs	Atrazine	Colorimetric and SERS	0.005 mg/L to 1 mg/L	<0.01 mg/L (Colorimetric) 0.0012 mg/L (SERS)	[33]
5	Methacrylic acid, ethylene glycol dimethacrylate	Atrazine	Electrochemical	5 × 10^−7^ to 5 × 10^−6^ M	4 × 10^−7^ M	[34]
6	Poly(3,4-ethylenedioxy thiophene-co-3-thiophene acetic acid	Atrazine	Electrochemical	10^−8^ to 10^−4^ mol/L	10^−9^ mol/L	[35]
7	Methacrylic acid, ethylene glycol dimethacrylate	Atrazine, propazine, terbuthylazine, des-ethyl atrazine	Mass sensitive	0.35 mg/L to 7 mg/L	0.1 to 0.2 mg/L for ATR, PRO, and TBA 0.1 mg/L or less for DEA	This Study

## Data Availability

The original contributions presented in the study are included in the article, further inquiries can be directed to the corresponding author.

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
