# Peer review of "Sensitive Coatings Based on Molecular-Imprinted Polymers for Triazine Pesticides’ Detection†"

_sensors, 2024, doi:10.3390/s24185934_

Round 1

Reviewer 1 Report (New Reviewer)

Comments and Suggestions for Authors

Review Report

Manuscript: Molecular Imprinted Polymers-Based Sensitive Coatings for Triazine Pesticides Detection: Optimization and Application
Journal: Sensors MDPI

The manuscript addresses a significant and timely topic, emphasizing both cost savings and health/environmental benefits. While the authors have made substantial efforts, several revisions are necessary to enhance the manuscript's suitability for publication in Sensors.

Outlined below are the recommended revisions:

  1. Title: The term "optimization" refers to a set of mathematical principles and methods for solving quantitative problems, which does not align with the context of this study. It is advisable to reconsider the use of "optimization" in the manuscript title. Furthermore, the MAA/EGDMA ratio data presented in Figure 2 does not constitute an optimization method. Therefore, removing "optimization" from the title is recommended.

  2. Abstract: To make the abstract more informative, consider including specific findings, such as the optimal acid ratios identified for maximizing sensitivity in the sensor coating design and the corresponding frequency shift/layer height.

  3. Keywords: Adding more keywords, such as "pesticide," "MIP," and "acid-to-cross-linker ratio," could improve the visibility of the study.

  4. Redundancy: The following sentences convey the same message, so it's unnecessary to repeat them:

    a. "Excessive use of pesticides is a major cause of contamination in agricultural crops [4]."
    b. "Excessive and prolonged use of pesticides beyond recommended levels has raised significant concerns about food safety and environmental health."

  5. Limitations: While the importance of molecularly imprinted polymers (MIPs) is highlighted, it is recommended to discuss the limitations and challenges associated with them.

  6. Introduction: The statement "In this study, the acid-to-cross-linker ratio was optimized to develop a stable polymeric system for effective recognition as well as re-inclusion of templated analyte" would benefit from a more detailed explanation in the introduction, covering the concept and significance of the acid-to-cross-linker ratio.

  7. Pesticide Concentration: Clarification is needed on the criteria used for exposing the sensitive coating of pyrrolidone to 8.6 mg/L of pesticide. Why was this specific concentration chosen?

  8. Ratio Justification: The manuscript lacks sufficient explanation for why the 46% MAA and 54% EGDMA ratio achieved higher values compared to other ratios. It is recommended to support this with references.

  9. Temperature Range: Effective sensor design should accommodate a broad range of field temperatures, which can reach up to 45°C in some regions. Relying on a single experimental temperature (28°C) may compromise the results' reliability.

  10. Humidity Effects: The study does not discuss the effect of relative humidity on the formation of the coating layer or the interactions between the components.

  11. Conclusion: Conclusions should summarize the key findings of the study, supported by relevant data. Referring to a specific figure (Figure 9) and reference [40] in the conclusions section should be avoided.

Comments on the Quality of English Language

The manuscript needs a simple linguistic review, especially in the use of the articles (the, a), in addition to the necessity of using "," before which in many cases. When mentioning figures and tables in the manuscript text, use the first letter as uppercase.

Author Response

We greatly appreciate the comments and suggestions from the reviewers. The points raised by reviewers have been addressed. These revisions are clearly highlighted in the manuscript. The detail of revisions is given here too. 

The manuscript addresses a significant and timely topic, emphasizing both cost savings and health/environmental benefits. While the authors have made substantial efforts, several revisions are necessary to enhance the manuscript's suitability for publication in Sensors.

Outlined below are the recommended revisions:

  1. Title: The term "optimization" refers to a set of mathematical principles and methods for solving quantitative problems, which does not align with the context of this study. It is advisable to reconsider the use of "optimization" in the manuscript title. Furthermore, the MAA/EGDMA ratio data presented in Figure 2 does not constitute an optimization method. Therefore, removing "optimization" from the title is recommended.

Ans: As recommended, the title of the manuscript has been revised.

  1. Abstract: To make the abstract more informative, consider including specific findings, such as the optimal acid ratios identified for maximizing sensitivity in the sensor coating design and the corresponding frequency shift/layer height.

Ans: As suggested, the abstract now includes the following information: A maximum sensor response of 546 Hz (frequency shift/layer height = 87.36) was observed for a sensitive coating composed of 46% methacrylic acid and 54% ethylene glycol dimethacrylate, with a demonstrated layer height of 250 nm (6.25 kHz).

  1. Keywords: Adding more keywords, such as "pesticide," "MIP," and "acid-to-cross-linker ratio," could improve the visibility of the study.

Ans: The suggested keywords have been incorporated into the manuscript.

  1. Redundancy: The following sentences convey the same message, so it's unnecessary to repeat them:
  1. "Excessive use of pesticides is a major cause of contamination in agricultural crops [4]."
    "Excessive and prolonged use of pesticides beyond recommended levels has raised significant concerns about food safety and environmental health."

Ans: It has been addressed, and the repeated sentence have been removed from the introduction.

  1. Limitations: While the importance of molecularly imprinted polymers (MIPs) is highlighted, it is recommended to discuss the limitations and challenges associated with them.

As recommended, limitations are also discussed in introduction section such as: MIPs hold significant promise for selective recognition across various applications. However, their practical use is hindered by several challenges, including incomplete template removal, limited availability of templates, heterogeneity of binding sites, and issues with mechanical and chemical stability under harsh conditions. Additionally, slow kinetics and limited reusability can negatively impact the reproducibility and selectivity of MIPs

  1. Introduction: The statement "In this study, the acid-to-cross-linker ratio was optimized to develop a stable polymeric system for effective recognition as well as re-inclusion of templated analyte" would benefit from a more detailed explanation in the introduction, covering the concept and significance of the acid-to-cross-linker ratio.

Ans: In designing a MIP using methacrylic acid (MAA) as the functional monomer and ethylene glycol dimethacrylate (EGDMA) as the crosslinker, the ratio of acid to crosslinker is a crucial factor that influences the polymer's binding efficiency, selectivity, and mechanical properties. The optimal ratio often depends on the nature of the template molecule and the strength of the interaction between MAA and the template. A higher amount of meth-acrylic acid can increase the number of available binding sites whereas a higher proportion of EGDMA increases the crosslinking density, resulting in a more rigid and stable polymer matrix. This rigidity is beneficial for maintaining the shape and integrity of the binding sites. The choice of solvent and polymerization conditions also influence the optimal monomer-to-crosslinker ratio, as these factors affect the polymer's porosity, surface area, and overall performance. Numerous studies in literature have designed Molecular Imprinted Polymers (MIPs) by optimizing the MAA to EGDMA ratio.

  1. Pesticide Concentration: Clarification is needed on the criteria used for exposing the sensitive coating of pyrrolidone to 8.6 mg/L of pesticide. Why was this specific concentration chosen?

Ans: A sensitive coating based on pyrrolidone was synthesized using 8.6 mg/L of pesticide as a template. This coating was then exposed to propazine concentrations of 2.1, 4.3, and 8.6 mg/L, with the highest response of 350 Hz observed at 8.6 mg/L of propazine.

  1. Ratio Justification: The manuscript lacks sufficient explanation for why the 46% MAA and 54% EGDMA ratio achieved higher values compared to other ratios. It is recommended to support this with references.

Ans: In designing a MIP using methacrylic acid (MAA) as the functional monomer and ethylene glycol dimethacrylate (EGDMA) as the crosslinker, the ratio of acid to crosslinker is a crucial factor that influences the polymer's binding efficiency, selectivity, and mechanical properties. The optimal ratio often depends on the nature of the template molecule and the strength of the interaction between MAA and the template. A higher amount of meth-acrylic acid can increase the number of available binding sites whereas a higher proportion of EGDMA increases the crosslinking density, resulting in a more rigid and stable polymer matrix. This rigidity is beneficial for maintaining the shape and integrity of the binding sites. The choice of solvent and polymerization conditions also influence the optimal monomer-to-crosslinker ratio, as these factors affect the polymer's porosity, surface area, and overall performance. Numerous studies in literature have designed Molecular Imprinted Polymers (MIPs) by optimizing the MAA to EGDMA ratio.

  1. Temperature Range: Effective sensor design should accommodate a broad range of field temperatures, which can reach up to 45°C in some regions. Relying on a single experimental temperature (28°C) may compromise the results' reliability.

Ans: Of course, a minor temperature effect on thermodynamics of analyte incorporation exists. But analytical challenge will be near to temperature chosen.

  1. Humidity Effects: The study does not discuss the effect of relative humidity on the formation of the coating layer or the interactions between the components.

Ans: Synthesis of coating was performed under anhydrous condition. The sensors were allowed to be equilibrated in aqueous solution before measurement!

  1. Conclusion: Conclusions should summarize the key findings of the study, supported by relevant data. Referring to a specific figure (Figure 9) and reference [40] in the conclusions section should be avoided.

Ans: As suggested, references to specific figures and citations have been removed from the conclusion. 

Reviewer 2 Report (New Reviewer)

Comments and Suggestions for Authors

--Title --Molecular imprinted polymers based sensitive coatings for tria- 2 zine pesticides detection: optimization and application (which application)!

-- Explain more about Figure 3.

--Why this shape of AFM in Figure 4.  like this?

--In table 1 your result is good or not and why

Author Response

We greatly appreciate the comments and suggestions from the reviewers. The points raised by reviewers have been addressed. These revisions are clearly highlighted in the manuscript. The detail of revisions is given here too.

Comments and Suggestions for Authors

--Title --Molecular imprinted polymers based sensitive coatings for triazine pesticides detection: optimization and application (which application)!

Ans: The title of the manuscript has been changed.

-- Explain more about Figure 3.

Ans: Additional explanation has been included as suggested.

--Why this shape of AFM in Figure 4.  like this?

Ans: The surface roughness of the imprinted layer coated on the QCM electrode was analyzed using an Atomic Force Microscope (AFM). Figure 4 shows the AFM image of the molecular imprinted polymer coating, captured in contact mode with a silicon nitride tip, revealing the surface morphology of the pesticide recognition layer.

--In table 1 your result is good or not and why

Ans: The limit of detection (LOD) for the developed sensor based on QCM is considered effective at 0.1 mg/L or lower, given the mass-sensitive transduction principle. The LOD can be significantly improved by using devices that operate at higher frequencies, such as in the GHz range.

Reviewer 3 Report (New Reviewer)

Comments and Suggestions for Authors

The authors report on the development of sensors for pesticide detection, based on molecular imprinting. They consider 4 pesticides and find that each sensitive coating is selective towards its templated analyte, except for propazine.

What is the reason that propazine is not selective towards its templated analyte?

The title could be misread as a study on polymers. "MIP-based sensitive coatings" or "Sensitive coating based on molecular-imprinted polymers" might be less ambiguous.

Introductory sentences like 12-14 do not belong in an abstract. Likewise acronym definitions (DEA,PRO, TBA, MIP, QCM) would be better stated in the main text. Some acronyms are defined multiple times (e.g. "des-ethyl atrazine (DEA)" appears 11 times), which defeats the purpose. Other acronyms remain undefined (HPLC, GC-MS, NIP), whereas some definitions only appear in the conclusion section (QCM, LOD). Long chemical names could be abbreviated (e.g. "ethylene glycol dimethacrylate" appears 10 times). Readability would be improved by a single definition at first point of usage, in the main text.

Figure 2 shows the frequency-shift divided by the layer-height, but no units are given. It appears that layer height is converted to kHz, using that "1 kHz change in frequency is equal to 40 nm". What is the origin of this calibration? Since frequency-shift is measured in Hz, this ratio has units Hz/kHz, or 1/1000, i.e. the dimensionless values are out by this factor. This could be explained better. The data might be better presented on a xy-chart with fraction MAA on the x-axis; there is no need for a third dimension. It would be of interest to know the layer-height for each MAA fraction, to assess what is driving the sensor response. 

Would the optimal coating composition found for chlorotriazine, be different for the other pesticides studied? The conclusion could mention this finding, giving it was the objective of this study (lines 67-68).

Figure 9 shows the "normalized" sensor response. How is the normalization performed? According to the caption "Frequency shifts were normalized for layer thickness and frequency shifts of reference electrodes were subtracted in each case". Does it correspond to the sensor response labelled "Difference" in Figure 5-8, divided by the layer thickness converted to kHz? If so, which concentration was used? Does a normalized sensor response of 1 have a significance? 

Table 1, "Sr. No." (meaning?) 2 refers to Atrazine as ATZ. 

Table 1, the LOD of "Sr. No." 7 states two ranges: "0.1 to 0.2 mg/L" and "0.1 mg/L or less". Which range applies to which pesticide?

Comments on the Quality of English Language

Typos

Line 39: later-on: lateron (subsequently)

Line 51/52: alzheimer/parkinson: named deceases are usually capitalized

Line 106: that's why: that is why (for this reason/therefore)

Line 220: distill: distilled

Line 270: signal/noise ration: signal/noise ratio

Author Response

We greatly appreciate the comments and suggestions from the reviewers. The points raised by reviewers have been addressed. These revisions are clearly highlighted in the manuscript. The detail of revisions is given here too.

Comments and Suggestions for Authors

The authors report on the development of sensors for pesticide detection, based on molecular imprinting. They consider 4 pesticides and find that each sensitive coating is selective towards its templated analyte, except for propazine.

What is the reason that propazine is not selective towards its templated analyte?

The title could be misread as a study on polymers. "MIP-based sensitive coatings" or "Sensitive coating based on molecular-imprinted polymers" might be less ambiguous.

Ans: As suggested, the title of the manuscript has been revised.

Introductory sentences like 12-14 do not belong in an abstract. Likewise acronym definitions (DEA,PRO, TBA, MIP, QCM) would be better stated in the main text. Some acronyms are defined multiple times (e.g. "des-ethyl atrazine (DEA)" appears 11 times), which defeats the purpose. Other acronyms remain undefined (HPLC, GC-MS, NIP), whereas some definitions only appear in the conclusion section (QCM, LOD). Long chemical names could be abbreviated (e.g. "ethylene glycol dimethacrylate" appears 10 times). Readability would be improved by a single definition at first point of usage, in the main text.

Ans: Introductory sentences and acronyms have been removed from the abstract. Additionally, redundant acronym definitions have been reduced, while definitions for previously undefined acronyms have been added to the manuscript.

Figure 2 shows the frequency-shift divided by the layer-height, but no units are given. It appears that layer height is converted to kHz, using that "1 kHz change in frequency is equal to 40 nm". What is the origin of this calibration? Since frequency-shift is measured in Hz, this ratio has units Hz/kHz, or 1/1000, i.e. the dimensionless values are out by this factor. This could be explained better. The data might be better presented on a xy-chart with fraction MAA on the x-axis; there is no need for a third dimension. It would be of interest to know the layer-height for each MAA fraction, to assess what is driving the sensor response. 

Ans: To enhance clarity, Figure 2 has been replaced with a table that presents the data, highlighting the composition of the sensitive coatings, their respective layer heights, and net frequency shifts.

Would the optimal coating composition found for chlorotriazine, be different for the other pesticides studied? The conclusion could mention this finding, giving it was the objective of this study (lines 67-68).

Ans: The findings of this study has been incorporated in the conclusion section: The optimized sensitive coating composition includes 46% methacrylic acid and 54% ethylene glycol dimethacrylate, with a layer height of 250 nm.

Figure 9 shows the "normalized" sensor response. How is the normalization performed? According to the caption "Frequency shifts were normalized for layer thickness and frequency shifts of reference electrodes were subtracted in each case". Does it correspond to the sensor response labelled "Difference" in Figure 5-8, divided by the layer thickness converted to kHz? If so, which concentration was used? Does a normalized sensor response of 1 have a significance? 

Ans: All responses were adjusted to the same coating height, the most significant signal was taken as “1”. Furthermore, only specific sensor effects were considered. Unspecific NIP responses are minor than those of MIP signals. A successful imprinting procedure as in our case leads to NIP signals of perhaps 10-20% of the overall MIP responses. The responses in Fig. 8(new) are differences of MIP and NIP signal, only specific responses were considered. 

Table 1, "Sr. No." (meaning?) 2 refers to Atrazine as ATZ. 

Ans: The acronym for atrazine in the main text has been designated as ATR, and this update has also been reflected in the table.

Table 1, the LOD of "Sr. No." 7 states two ranges: "0.1 to 0.2 mg/L" and "0.1 mg/L or less". Which range applies to which pesticide?

Ans: It has now been correctly indicated in the table.

Comments on the Quality of English Language

Typos

Line 39: later-on: lateron (subsequently)

Line 51/52: alzheimer/parkinson: named deceases are usually capitalized

Line 106: that's why: that is why (for this reason/therefore)

Line 220: distill: distilled

Line 270: signal/noise ration: signal/noise ratio

Ans: These typos have now been corrected in the manuscript.

Round 2

Reviewer 1 Report (New Reviewer)

Comments and Suggestions for Authors

The authors have made significant improvements to the manuscript, effectively addressing most of the first review's concerns. However, they fail to provide a compelling explanation for the higher values of the 46% MAA and 54% EGDMA ratio, why these ratios yielded superior results, or if other previous studies can corroborate the research results. The authors' explanation for this particular point is not convincing.

Author Response

Dear Editor,

We greatly appreciate the comments and suggestions from the reviewer. The points raised by the reviewer have been addressed. These revisions are clearly highlighted in the manuscript. The detail of revisions is given here too with proper line numbers.

Sincerely,

F.L. Dickert and U. Latif

Open Review

Quality of English Language

( ) I am not qualified to assess the quality of English in this paper.
( ) The English is very difficult to understand/incomprehensible.
( ) Extensive editing of English language required.
( ) Moderate editing of English language required.
( ) Minor editing of English language required.
(x) English language fine. No issues detected.

Yes

Can be improved

Must be improved

Not applicable

Does the introduction provide sufficient background and include all relevant references?

(x)

( )

( )

( )

Is the research design appropriate?

(x)

( )

( )

( )

Are the methods adequately described?

(x)

( )

( )

( )

Are the results clearly presented?

( )

(x)

( )

( )

Are the conclusions supported by the results?

(x)

( )

( )

( )

Comments and Suggestions for Authors

The authors have made significant improvements to the manuscript, effectively addressing most of the first review's concerns. However, they fail to provide a compelling explanation for the higher values of the 46% MAA and 54% EGDMA ratio, why these ratios yielded superior results, or if other previous studies can corroborate the research results. The authors' explanation for this particular point is not convincing.

Ans: pg 4, lines 160-175

All sensor responses in our experiment are attributed to bulk phenomena. This conclusion is supported by test measurements where the layer height was increased without modifying other parameters. To further validate our findings, we performed comparative measurements using uncoated gold electrodes. Additionally, we employed the thiol compound (octadecanethiol) to create a dense monolayer on the gold electrode surface. This particular thiol was selected due to its molecular weight being comparable (even higher) to that of atrazine.

In our tests, the thiol produced a sensor response of 140 Hz, which is significantly lower than the 546 Hz response generated by Atrazine. This difference highlights the higher sensitivity of the sensor to atrazine confirming bulk effect.

The observed maximum in the MAA/EGDMA ratio is driven by two opposing mechanisms. On one hand, an increase in MAA enhances the sensor's interaction with atrazine, extending the binding affinity. On the other hand, a reduction in EGDMA results in fewer cavities within the polymer matrix, impacting its structural characteristics and vice versa. Therefore, it can be inferred that a host-guest interaction mechanism plays a pivotal role in the sensor response observed in this system.  

This manuscript is a resubmission of an earlier submission. The following is a list of the peer review reports and author responses from that submission.

Round 1

Reviewer 1 Report

Comments and Suggestions for Authors

The study "Molecular imprinted polymers based sensitive coatings for pesticide detection" developed sensors for detecting pesticides using thin films created through molecular imprinting technology. The study also investigated the interaction of triazine molecules with the hydrophobic core. The sensors demonstrated reversible sensory reactions.

Notes for correction:
The presentation of the work is clear and distinct, with clear and informative graphs and formulas. However, the conclusions are lacking in detail and could be expanded, particularly regarding the quantitative parameters and characteristics of the produced sensors.

Design of the work text:

-The work is written clearly and distinctly.

-Graphs and formulas are clear and informative.

In conclusion, the article "Molecular imprinted polymers based sensitive coatings for pesticide detection" can be published after minor corrections.

Author Response

Reviewer 1:

Thank you very much for the stimulating comments!

Moderate editing of English language required – all reviewers

Answer: We regret there were problems with the English. Now, the manuscript has been carefully revised to improve the grammar and readability.

The presentation of the work is clear and distinct, with clear and informative graphs and formulas. However, the conclusions are lacking in detail and could be expanded, particularly regarding the quantitative parameters and characteristics of the produced sensors.

Answer: Line numbers 306-321: The fabricated sensor devices exhibit selective reversible response towards their templated analytes while exposing to the concentration range from 0.35 mg/L to 7 mg/L. The propazine-imprinted sensitive coating-based sensor exhibits LOD of 0.21 mg/L, DEA-imprinted sensitive coating-based sensor exhibits LOD of 0.12 mg/L, TBA-imprinted sensitive coating-based sensor exhibits LOD of 0.12 mg/L and ATR-imprinted sensitive coating-based sensor exhibits LOD of 0.21 mg/L. Each sensor demonstrates higher sensitivity towards its specific templated analyte, showcasing how the imprinting technique can make similar polymers selective for a target analyte. This allows the selectivity of the sensor coating to be tailored to the analyte of interest. Each sensitive coating is selective and sensitive to its templated analyte. However, in the case of DEA, the sensor exhibits higher sensitivity to its templated analyte compared to the sensitivity of other coatings (PRO, TBA, ATR) to their respective analytes. The metabolites of Atrazine, namely Des-isopropyl atrazine (DIA) and Des-ethyl atrazine (DEA), have a pronounced basic nature, providing additional sites for hydrogen bonding with the acidic moieties of the polymer chain. These characteristics enable DEA-imprinted coatings to exhibit higher sensitivity compared to other coatings.

Reviewer 2 Report

Comments and Suggestions for Authors

In the paper by Latif et al. molecular imprinted polymers based sensitive coatings for pesticide detection were prepared and characterized.

The paper results interesting and well structured but needs further improvements to be accepted for publication.

The authors report at the end of the introduction section : "The combination of molecular imprinting polymers with QCM transducers enable us to develop highly sensitive and selective sensor for detecting pesticides such as atrazine (ATR) as well as its metabolites des-ethyl atrazine (DEA), des-isopropyl atrazine (DIA) and des-ethyl-des-isopropyl-atrazine (DEDIA) and structural analogues e.g. Simazine (SIM), Propazine (PRO) and Terbuthylazine (TBA)"

It is not clear how the syntetic approach is novel and which is the state of the art of this specific polymeric sensor.

How the results obtained with a target pesicide can be compared to the other conventional/non conventional sensors?

An in depth morphological characterization can be reported i.e. SEM detection together with informations about the layer deposition techniques, porosity of the sensor film.

Comments on the Quality of English Language

Minor English editing are required

Author Response

Reviewer 2:

Thank you very much for the stimulating comments!

Moderate editing of English language required – all reviewers

Answer: We regret there were problems with the English. Now, the manuscript has been carefully revised to improve the grammar and readability.

The paper results interesting and is well structured but needs further improvements to be accepted for publication.

The authors report at the end of the introduction section: "The combination of molecular imprinting polymers with QCM transducers enable us to develop highly sensitive and selective sensor for detecting pesticides such as atrazine (ATR) as well as its metabolites des-ethyl atrazine (DEA), des-isopropyl atrazine (DIA) and des-ethyl-des-isopropyl-atrazine (DEDIA) and structural analogues e.g. Simazine (SIM), Propazine (PRO) and Terbuthylazine (TBA)".

It is not clear how the synthetic approach is novel and which is the state of the art of this specific polymeric sensor.

Answer: (Line numbers 63-70): The strong yet reversible interaction between basic chlorotriazines and the acidic nature of the polymeric chain is a critical factor in designing a sensitive coating for the detection of target analyte (pesticide). In this study, the acid-to-cross-linker ratio was optimized to develop a stable polymeric system for effective recognition as well as re-inclusion of templated analyte.

How the results obtained with a target pesticide can be compared to the other conventional/nonconventional sensors?

Answer: These sensors are based on highly cross linked polymers (MIPs) which are very robust. Furthermore, mass sensitive sensors, e.g. QCMs show responses to all analytes adhered at the sensitive coatings. 

An in depth morphological characterization can be reported i.e. SEM detection together with informations about the layer deposition techniques, porosity of the sensor film.

Answer: (Line numbers 171-177) AFM image of the molecular imprinted polymer coating was taken in contact mode with silicon nitride tip, which revealed the surface morphology of the pesticide-recognition layer. TEM is most powerful tool among microscopic techniques [39], however, in this study, AFM was preferred over SEM or TEM for surface profiling because surface characteristics can be analyzed at ambient temperature than at high vacuum [37]. Furthermore, AFM can excellently characterize homogeneity of surface which is important to reduce noise of sensor responses.

Reviewer 3 Report

Comments and Suggestions for Authors

After revise the manuscript entitled: “Molecular imprinted polymers based sensitive coatings for pesticide detection: optimization and application” by Usman Latif, Sadaf Yaqub, and Franz L. Dickert, I consider it non-suitable for publication in Sensors. Despite the topic is interesting, the manuscript is not addressing any novelty in the experimental design or new insight about the improvement of using this technology. Several deficiencies in the methodology were detected and the characterization procedures were not discussed and the conclusions are not supported by the experimental results. For instance, scale bar in the AFM image was not labelled and the roughness were not given in distance units (nm, cm, m). The followed methodology is not clear and the employed materials were not mentioned in the “Materials and Methods”.

Comments on the Quality of English Language

Moderate editing of English language required

Author Response

Reviewer 3:

Thank you very much for the stimulating comments!

Moderate editing of English language required – all reviewers

Answer: We regret there were problems with the English. Now, the manuscript has been carefully revised to improve the grammar and readability.

After revise the manuscript entitled: “Molecular imprinted polymers based sensitive coatings for pesticide detection: optimization and application” by Usman Latif, Sadaf Yaqub, and Franz L. Dickert, I consider it non-suitable for publication in Sensors. Despite the topic is interesting, the manuscript is not addressing any novelty in the experimental design or new insight about the improvement of using this technology. Several deficiencies in the methodology were detected and the characterization procedures were not discussed and the conclusions are not supported by the experimental results. For instance, scale bar in the AFM image was not labelled and the roughness were not given in distance units (nm, cm, m). The followed methodology is not clear, and the employed materials were not mentioned in the “Materials and Methods”.

The scale bar in the AFM image is now properly labelled. The methodology which was followed to develop MIP-based gravimetric sensors is already discussed in our previous study [38]. Moreover, the material which were used to develop pesticides sensors are now mentioned in the “Materials and Methods” section.

Round 2

Reviewer 2 Report

Comments and Suggestions for Authors

The paper can be accepted for publication

Comments on the Quality of English Language

Minor English editing is required

Reviewer 3 Report

Comments and Suggestions for Authors

Dear authors,

In my opinion, the modifications made to to manuscript do not reach the treshold for a new revision. I decide to reject the manuscript for its publication in Sensors. I understand that the methodology could be a modification of a previous work, but it must be well explained or at least well referenced. 

Comments on the Quality of English Language

Minor editing of English language required